

# The extremely hot and dry 2018 summer in central and northern Europe from a multi-faceted weather and climate perspective

Efi Rousi[1], Andreas H. Fink[2], Lauren S. Andersen[1], Florian N. Becker[2], Goratz Beobide-Arsuaga[3,4], Marcus Breil[2,5], Giacomo Cozzi[6,7], Jens Heinke[1], Lisa Jach[8], Deborah Niermann[9], Dragan Petrovic[10], Andy Richling[11], Johannes Riebold[12], Stella Steidl[9], Laura Suarez-Gutierrez[13], Jordis Tradowsky[6,14], Dim Coumou[1,15,16], André Düsterhus[17], Florian Ellsäßer[18], Georgios Fragkoulidis[19], Daniel Gliksman[20,21], Dörthe Handorf[12], Karsten Haustein[22,#], Kai Kornhuber[1,23,24], Harald Kunstmann[7,10], Joaquim G. Pinto[2], Kirsten Warrach-Sagi[8], Elena Xoplaki[18,25]

[1]Potsdam Institute of Climate Impact Research (PIK), Member of the Leibniz Association, Potsdam, Germany

[2]Institute of Meteorology and Climate Research (IMK-TRO), Karlsruhe Institute of Technology, Karlsruhe, Germany

[3]International Max Planck Research School on Earth System Modelling (IMPRS-ESM), Germany

[4]Institute of Oceanography, Center for Earth System Sustainability, Hamburg University, Hamburg, Germany

[5]University of Hohenheim, Hohenheim, Germany

[6]Deutscher Wetterdienst, Regionales Klimabüro Potsdam, Stahnsdorf, Germany,

[7]University of Augsburg, Augsburg, Germany

[8]University of Hohenheim, Hohenheim, Germany

[9]Deutscher Wetterdienst, Offenbach, Germany

[10]Institute of Meteorology and Climate Research (IMK-IFU), Karlsruhe Institute of Technology, Campus Alpin, Garmisch-Partenkirchen, Germany

[11]Institute of Meteorology, Free University of Berlin, Berlin, Germany

[12]Alfred Wegener Institute, Helmholtz Centre for Polar and Marine Research, Potsdam, Germany

[13]Max-Planck-Institut für Meteorologie, Hamburg, Germany

[14]Bodeker Scientific, Alexandra, New Zealand

[15]IVM-Institute for Environmental Studies, Free University of Amsterdam, Amsterdam, Netherlands

[16]Royal Netherlands Meteorological Institute (KNMI), De Bilt, Netherlands

[17]Irish Climate Analysis and Research UnitS (ICARUS), Department of Geography, Maynooth University, Maynooth, Ireland

[18]Centre of International Development and Environmental Research, Justus Liebig University Giessen, Giessen, Germany

[19]Institute for Atmospheric Physics, Johannes Gutenberg University, Mainz, Germany

[20]Institute for Hydrology and Meteorology, Faculty of Environmental Sciences, Technische Universität Dresden, Tharandt, Germany

[21]Institute of Geography, Technische Universität Dresden, Dresden, Germany





[22]Climate Service Center Germany (GERICS), Helmholtz-Zentrum hereon, Hamburg, Germany
[23]Lamont-Doherty Earth observatory, Columbia University, New York, US
[24]German Council on Foreign Relations, Berlin, Germany
[25]Institute of Geography, Justus Liebig University Giessen, Giessen, Germany
# Now at Institute for Meteorology, University of Leipzig, Leipzig, Germany
*Correspondence to*: Efi Rousi (rousi@pik-potsdam.de) and Andreas H. Fink (andreas.fink@kit.edu)
**Abstract.** The summer of 2018 was an extraordinary season in climatological terms for northern and
central Europe, bringing simultaneous, widespread, and concurrent heat and drought extremes in large
parts of the continent with extensive impacts on agriculture, forests, water supply, and socio-economic
sector. We present a comprehensive, multi-faceted analysis of the 2018 extreme summer in terms of
heat and drought in central and northern Europe with a particular focus on Germany. The heatwave first
affected Scandinavia by mid-July, shifted towards central Europe in late July, while Iberia was primarily
affected in early August. The atmospheric circulation was characterized by strongly positive blocking
anomalies over Europe, in combination with a positive summer North Atlantic Oscillation and a double
jet stream configuration before the initiation of the heatwave. In terms of possible precursors common
to previous European heatwaves, the Eurasian double jet structure and a tripolar sea-surface temperature
anomaly over the North Atlantic were identified already in spring. While in the early stages over
Scandinavia the air masses at mid- and upper-levels were often of remote, maritime origin, at later
stages over Iberia the air masses had primarily a local to regional origin. The drought affected Germany
the most, starting with warmer than average conditions in spring, associated with enhanced latent heat
release that initiated a severe depletion of soil moisture. During summer, a continued precipitation
deficit exacerbated the problem, leading to hydrological and agricultural drought. A probabilistic
attribution assessment of the heatwave in Germany showed that the prolonged heat has become more
likely due to global warming. Regarding future projections, an extreme summer such as this of 2018 is
expected to occur every two out of three years in Europe under a 1.5 °C warmer world and virtually
every single year under 2 °C of global warming. With such large-scale and impactful extreme events
becoming more frequent and intense under anthropogenic climate change, comprehensive and multi-
faceted studies like the one presented here quantify the multitude of effects and provide valuable
information as basis for adaptation and mitigation strategies.

## 1   Introduction

Following an anomalously warm and dry spring, the summer of 2018 was characterized by record
breaking widespread heat and drought across Europe (Kennedy et al., 2019; Toreti et al., 2019) with
intense heatwaves affecting large parts of Scandinavia (Sinclair et al., 2019) and central Europe (e.g.,



Vogel et al. 2019). In Germany, both the months of April-May, as well as the April-July period, and the
entire year, were identified as the warmest in the observational records starting in 1881. Moreover,
Germany faced remarkably prolonged drought from February to November, with 2018 being the fourth
driest year on record (after 1959, 1911, and 1921). A new record was also set for annual sunshine
duration, amounting to 2015 hours (Friedrich and Kaspar, 2019). In Finland, the peak temperature in
summer exceeded 33 °C, which is extremely unusual for a region located near the Arctic Circle,
breaking historical records of the past 40 years (Liu et al., 2020). In the UK, summer 2018 joined 2006
as the hottest on record since 1884. In England itself, this was the warmest on record, while June 2018
was the driest June for England since 1925 (Kendon et al., 2019). Over the Iberian peninsula, a heatwave
developed in early August 2018, with this month being the warmest in the region after 2003
(Barriopedro et al., 2020). The normal eastward propagation of weather systems was hindered in
summer 2018 by the recurrent presence of blocking anticyclones, associated with a particularly
meandering jet stream, which was reflected in the way the heatwave propagated, starting in Scandinavia
(peaking mid of July), then developing in central Europe (end of July) and last in Iberia (beginning of
August). For the European continent, 2018 was the second warmest summer on record (following 2010)
as estimated from the CRUTEM4 dataset (Kennedy et al., 2019), prior to being marginally surpassed
by the 2021 summer (Copernicus Climate Change Service, 2018; 2021).

In terms of amplitude, persistence and spatial extent, the 2018 heatwaves were comparable to the
"mega-heatwaves" of 2003 and 2010 over Europe and Russia (Spensberger et al., 2020; Becker et al.,
2022), during which more than one million square kilometers were simultaneously affected by heatwave
conditions (Fink et al., 2004; Barriopedro et al., 2011). But, unlike 2003 and 2010, the exceptionally
extreme heat in 2018 occurred under concurrent exceptionally dry conditions, thus making the events
in 2018 a spatially and temporally compound extreme (Zscheischler et al. 2020; Bastos et al. 2021;
Ionita et al., 2021). These co-occurring hot and dry extremes, not only in central Europe but also in
multiple regions of the northern hemisphere midlatitudes (Vogel et al., 2019), caused vast aggregated
impacts (Bakke et al., 2020), ranging from drought-inflicted forest mortality events of unprecedented
scale (Schuldt et al., 2020; Senf and Seidl, 2021), up to 50 % reduction in agricultural yields (Toreti et
al., 2019; Beillouin et al., 2020) and increased forest fire occurrence (San-Miguel-Ayanz et al., 2019),
to excess heat-related human mortality (Pascal et al., 2021). Compared to previous droughts since 2000,
summer 2018 occupied the largest extent of extreme and severe agriculture drought, centered around
Germany, Poland, most of Scandinavia and the Baltic countries, affecting a larger extent of boreal
forests and high latitude ecosystems (Peters et al., 2020). Further, from a temporal point of view,
compared to other droughts of the past 40 years, 2018 was characterized by the sharpest transition from
average-to-wet conditions in late winter to extremely strong soil-water deficits in summer (Bastos et
al., 2020).

Surface heatwaves are typically co-located with the center of the associated blocking system (Kautz
et al., 2022; their Figure 2b). If the blocking is intense and persistent, a heatwave will usually develop.



On the other hand, unsteady weather conditions, like thunderstorms and heavy precipitation, are
frequent on the flanks of the blocking system, which correspond to the air mass boundaries (Kautz et
al., 2022). In fact, summer extremes can be exacerbated by different components of the Earth system,
such as anomalous atmospheric circulation patterns, oceanic conditions, and the state of land surface
(Wehrli et al., 2019; Di Capua et al., 2021). The atmospheric circulation during late spring and summer
2018 was characterized by the frequent presence of atmospheric blocking, and a persistent positive
summer North Atlantic Oscillation (sNAO; Drouard et al., 2019; Li et al., 2020). Among the possible
precursors of European heatwaves, here we analyzed spring sea-surface temperatures (SST) over the
North Atlantic and soil moisture anomalies over Europe. In particular, the tripolar North Atlantic SST
anomaly pattern is known to be influenced by the winter NAO, persisting over spring and affecting
European climate in summer (Herceg-Bulić and Kucharski, 2014). The North Atlantic tripolar pattern
has been associated with the East Atlantic Pattern (Gastineau and Frankignoul, 2015) and Atlantic
Ridges (Ossó et al., 2020), leading to decreased summer precipitation (Saeed et al., 2013; Rousi et al.,
2021) and increased summer temperatures over Europe (Chen et al., 2016). Additionally, Duchez et al.
(2016) argues that a cold anomaly over the North Atlantic subpolar gyre (SPG) may be associated with
a stationary position of the jet stream, enhancing European summer heat extremes. Moreover, soil
moisture-temperature feedbacks can amplify heat extremes (Seneviratne et al., 2010). Through a
positive feedback, soil moisture depletion by hot and dry atmospheric conditions leads to a reduction
of evaporative cooling and suppressed convective available potential energy (CAPE) values,
subsequently limiting the rainfall potential and increasing air temperatures further (Miralles et al., 2014;
Prodhomme et al., 2021).
Hot and dry summers in Europe are expected to occur more frequently under anthropogenic global
warming (Masson-Delmotte et al. IPCC, 2021). McCarthy et al. (2019) conducted an attribution study
for the 2018 summer heatwave in the UK based on CMIP5 models and found that the present-day
likelihood of such extremes is around 11 %, which has been made 30-times higher due to anthropogenic
climate change, while this likelihood increases to 53 % by the 2050s.  Given the increase of hot and dry
extremes in Europe (Manning et al., 2019; Perkins-Kirkpatrick and Lewis, 2020; Markonis et al., 2021)
and their further expected increase under continued unmitigated anthropogenic climate change (Russo
et al., 2014; 2015; Spinoni et al., 2018; 2020), comprehensive weather and climate studies analyzing
regional heatwave and drought characteristics, drivers, and impacts are particularly important.
Within the German research initiative ClimXtreme, about 140 scientists from 35 institutions joined
in   39   projects   to   further   understand   climate   extremes,   focusing   on   central   Europe
(https://climxtreme.net/index.php/en/). Inter-disciplinary task forces were formed, among which one on
heat and drought. This study brings together its members to study the 2018 European heat and drought
from a multi-faceted weather and climate perspective, making it the first comprehensive study looking
at hot and dry summers over Europe using different analyses approaches to study (a) the extremeness
and attribution to anthropogenic climate change (climate perspective), as well as (b) the synoptic



dynamics in concert with the role of slowly varying boundary conditions at the ocean and continental
surfaces (seasonal and weather perspective). In the following, first, the data and methods are presented
(Sect. 2). Different metrics for the detection and description of the 2018 summer extremes are shown
in Sect. 3.1. Then, we present various features of the atmospheric circulation including blocking, jet
stream state, weather regimes, Rossby wave activity and air mass trajectories (see Sect. 3.2). Next, the
role of low-frequency precursors, i.e. SSTs and soil moisture in spring, in setting the scene and
eventually shaping those extremes is investigated (see Sect. 3.3). Sect. 3.4. examines the event from a
large-ensemble climate model perspective, accompanied by a tailored attribution analysis of the
heatwave in Germany based on CMIP6 models. The Discussion and Conclusions section completes this
paper.
**2 Data and Methods**
**2.1 Data**
In this paper we use a variety of datasets, including observational, reanalysis and model data. We are
using a common spatial domain for Europe (10° W–50 °E, 30°-70° N) and the reference period 1981–
2010 unless otherwise stated.
ERA5 (Hersbach et al., 2020) and ERA5-HEAT (Di Napoli et al., 2021) reanalysis datasets were
utilized for the calculation of heatwave metrics (see Sect. 3.1), the dynamical drivers and their evolution,
such as Rossby wave activity, backward trajectories, double jet streams, atmospheric blocking, and
weather regimes (see Sect. 3.2), as well as the precursors, i.e. SSTs and soil moisture (see Sect. 3.3). E-
OBS gridded observational datasets (Haylock et al., 2008; Cornes et al., 2018) were used for the
calculation of the drought indices (SPI, SPEI), for the drought detection with climate networks (see
Sect. 3.1), and to estimate the return period of the heatwave and select equivalent extreme events in
CMIP6 model simulations for the attribution study (see Sect. 3.4). Observational datasets from DWD
stations (Kaspar et al., 2013) were used for the thermopluviogram for Germany (see Sect. 3.1).
The dynamic vegetation model „LPJmL5-tillage" (Von Bloh et al., 2018; Schaphoff et al., 2018;
Lutz et al., 2019) was used to simulate soil moisture as forced by climate parameters (i.e. temperature,
precipitation, wind) from the GSWP3-W5E5 dataset, a combination of GSWP3 v1.09 (Kim, 2017) and
a bias-adjusted version of ERA5 reanalysis data (Lange, 2019). The simulation was run under evolving
$CO_2$ and pre-industrial natural vegetation conditions (see Sect 3.3).
The historical and RCP4.5 simulations of the Max Planck Grand Ensemble (MPI-GE; Maher et al.,
2019) were used to calculate the cumulative excess heat under recent climate (1979-2021), and future
1.5˚ C (2020-2049) and 2˚ C (2050-2079) warmer worlds (see Sect. 3.4). The advantage of this dataset
is that, apart from the forced response, it provides an estimate of the internal natural variability.
Historical simulations of several Coupled Model Intercomparison Project Phase 6 models (CMIP6;



Eyring et al., 2016) and pre-industrial type simulations (hist-nat) of the same models from the CMIP6-
endorsed Detection and Attribution Model Intercomparison Project (DAMIP; Gillett et al., 2016) were
used for the probabilistic attribution study (see Sect. 3.4). An overview of the analyzed CMIP6 models
is given in Table A1.
**2.2 Methods**
**2.2.1 Heatwave metrics**
Despite the fact that heatwaves have been a topic of active climate research for many decades, there is
no universal heatwave definition and there are multiple metrics and criteria depending on the region,
the season, and the purpose of the study (Becker et al., 2022). Here, we chose two different metrics to
characterize heatwave intensity, the cumulative heat, which uses temperature only, and the cumulative
Universal Thermal Climate Index (cUTCI) that represents human thermal comfort, taking into account
temperature, humidity, wind, and radiation.

A heatwave was defined here as an event of at least three consecutive days during which the 90[th]

percentile of the daily maximum temperature based on each calendar day is exceeded (Fischer and
Schär, 2010). Cumulative heat refers to the integration of heat exceedance over the threshold for all
heatwave days of a season. In the present study, only summer months  (June to August; JJA) were
considered, hence combining the intensity and persistence of heatwaves (Perkins-Kirkpatrick and
Lewis, 2020). The cUTCI was calculated for each day as in Błazejczyk et al. (2013) and the 90[th]
percentile of the daily time series was defined. The cumulative intensity was then calculated as the
integration of the exceedance above this threshold for the whole season.
**2.2.2 Drought indicators**
For the characterization and detection of the 2018 drought we used two widely accepted indicators, the
Standardized Precipitation Index (SPI; Mckee et al., 1993) and the Standardized Precipitation
Evapotranspiration Index (SPEI; Vicente-Serrano et al., 2014), and one alternative method based on
climate networks (Tsonis et al., 2006; Donges et al., 2009).

Two aggregation periods, three and six months, were selected, so that two types of droughts, could

be considered, meteorological (SPEI3) and agricultural (SPEI6) (Heim, 2002; Zampieri et al., 2017).
The SPEI was calculated with the SPEI R Package (Beguería and Vicente-Serrano, 2013). For the SPI,
monthly precipitation sums were used, while for the SPEI additionally monthly mean maximum and
minimum temperatures were needed for the calculation of the potential evapotranspiration (PET). This
was conducted based on the modified Hargreaves equation (Droogers and Allen, 2002). The method
corrects the PET calculated by the Hargreaves equation by using the monthly rainfall amount as a proxy
for insolation and based on the hypothesis that this amount can change the humidity levels (Vicente-





Serrano et al., 2014). The values obtained by this method are similar to those obtained from the Penman-
Monteith method (Allen et al., 2006).

Further, a climate network approach was used (Tsonis et al., 2006; Donges et al., 2009) to detect

drought conditions in Germany. In a climate network, high spatial coherence of weather conditions is
characterized by high values of the node degree measure that accounts for pairwise statistical similarity
(e.g. quantified with the Pearson correlation coefficient; for more details about the construction of the
climate network see Schädler and Breil, 2021). In this context, the node degree of a single grid point is
the number of network nodes (or grid points) connected to it. The higher the node degrees of a climate
network, the higher the spatial coherence of the meteorological time series and thus, the similarity of
the weather conditions. Since droughts are typically extensive and persistent events, high node degrees
can be used as a good drought indicator.

### 2.2.3 Atmospheric circulation metrics

The large-scale atmospheric circulation patterns and the dynamical evolution of the atmosphere
associated with the 2018 extremes were analyzed using various metrics. First, we looked at the weather
regimes during summer in order to characterize large-scale circulation features. Five summer
circulation regimes were computed with k-means clustering (Crasemann et al., 2017) applied to ERA5
sea-level pressure (SLP) anomalies for the time period 1979-2018 over the North Atlantic/European
region (30-88° N, 90° W - 90° E). Further, blocking frequency anomalies were calculated at a grid point
level based on a slightly modified version of the two-dimensional blocking index from Scherrer et al.
(2006). Daily blocked grid points were identified based on gradients in the daily 500 hPa geopotential
height (gph) field and on areas of positive gph anomalies associated with the blocking detection, as
described in Schuster et al. (2019).

Next, we looked at the state of the jet stream. Jet stream states were identified with the use of Self-

Organizing Maps (SOMs), a neural network-based clustering algorithm (Kohonen, 2013; Rousi et al.,
2015). SOMs were applied on daily ERA5 data of Eurasian (25-80° N, 25° W - 180° E) zonal-mean
zonal wind data on different pressure levels (800 hPa -100 hPa) for the time period 1979-2020 (see
details in Rousi et al., 2022). Moreover, we applied the methodology of Fragkoulidis and Wirth (2020)
to identify Rossby wave packets and their amplitude (E) for the 2018 summer. The method employs the
meridional wind field (v) at 300 hPa at 2x2 degree resolution, which was taken from the ERA5 data.
The visualization of E and v (see Fig. 4) is adaptive to the latitude location of strong Rossby wave
packets and only the latitudinal belt of 40-90° N was taken into account. For each longitude, E and v
are averaged over 10 grid points that exceed the median of all values within that belt.

To analyze the origin of the air masses during the 2018 summer heatwave, we calculated backward

trajectories using Lagrangian analysis and the LAGRANTO tool (Sprenger and Wernli, 2015). In
particular, we calculated 10-day backward trajectories for the levels between 1000 and 500 hPa in steps
of 25 hPa using ERA5 data for three starting locations in Europe on the respective peak heatwave days.





As in Zschenderlein et al. (2020), starting points were also taken within the upper-tropospheric blocking
anticyclone, in this case over Scandinavia. These were defined as the grid points where the anomaly of
the vertically averaged potential vorticity (between 500 and 150 hPa, based on monthly climatology)
was below -0.7 PVU (1 PVU = $10^{-6}$ K kg$^{-1}$ m² s$^{-1}$). For all grid points that fulfilled this criterion,
trajectories were initialized every 50 hPa between 500 and 150 hPa in the vertical dimension. To exclude
starting points in the stratosphere, only grid points with PV < 1 PVU were considered.
**2.2.4    Low-frequency precursors**
In order to analyze low-frequency precursors of the summer 2018 extremes, we considered SSTs, total
precipitation and soil moisture in the preceding months. The SST anomalies, compared to the reference
period of 1981-2010, over the North Atlantic and the seas surrounding Europe (Mediterranean, North
Sea, Baltic Sea) were analyzed for the spring (March to May; MAM) and summer (June to August;
JJA) months of 2018 in ERA5 data. Precipitation and soil moisture anomalies over Europe were also
calculated for the same seasons in ERA5, and for soil moisture LPJmL simulations were also used.
Additionally, we derived time series for the soil moisture-latent heat flux correlation in Germany
based on ERA5 reanalysis data and LPJml output with a daily temporal resolution based on centered
92-day running windows. This approach was used because soil moisture limitation depends on various
factors, such as the climatic conditions and vegetation characteristics (rooting depth, Leaf Area Index
(LAI) and stomatal conductance), which vary spatially and can change during the course of a year (Duan
et al., 2020). Therefore, the limitation cannot be easily represented by a unified, fixed value. The time
series were spatially averaged over all land points for Northern Germany and surroundings (51-55° N
and 4-16° E), as well as southern Germany and surroundings (48-51° N and 4-16° E). The German
Alpine region was not included in the southern German region because the complex topography that
cannot be accounted for in this study, influences the results.
**2.2.5 Attribution of the 2018 extreme heat**
Extreme event attribution typically addresses the question of whether and to what extent climate change
has affected the severity and/or frequency of a specific extreme weather event (Shepherd, 2016). The
most commonly used approach to extreme event attribution is probabilistic event attribution (Philip et
al., 2020), which compares climate model simulations under different scenarios, i.e. a factual scenario
which simulates the weather under current and past climate conditions, and a counterfactual scenario
which simulates weather under climate conditions excluding anthropogenic influences.
Here we present two kinds of attribution approaches. In the first, we used the MPI-GE to estimate
the probability of exceedance of the 2018 summer heat levels in the whole European domain for present
and future climates, and in the second, we present a tailored extreme event attribution study for
Germany based on CMIP6 simulations to calculate probability ratios for the persistent 2018 heat event
in Germany.



The MPI-GE (Maher et al., 2019) was used to estimate and compare the probabilities of exceeding
the 2018 summer levels of cumulative heat in the reanalysis data (ERA5, 1979–2021) and under recent
(1979–2021), and future 1.5° C (2020–2049) and 2° C (2050–2079) warmer worlds. The same heatwave
metric and parameters were used to calculate the cumulative heat as the ones described above (Sect.
2.2.1). The ERA5 data were regridded to a coarser resolution to match that of the MPI-GE and the
probabilities were normalized to percentages (i.e. divided by the total number of years in each period).
Then, to estimate how the occurrence probability of the 2018 heatwave in Germany has been affected
by anthropogenic climate change, a tailored probabilistic attribution study was conducted using CMIP6
simulations. The historical CMIP6 simulations provide the factual scenario while hist-nat simulations
from DAMIP provide the counterfactual scenario. The analysis is based on an attribution system
currently under development at DWD within the ClimXtreme project and involves (1) defining the
extreme event, (2) analyzing observational data and estimating the probability/return period of such an
event based on observations, (3) validating the climate model simulations, (4) preparing and analyzing
the climate model simulations, and (5) calculating a probability ratio between the historical and hist-nat
simulations.
Based on CMIP/DAMIP data available at the computing facility of the German Climate Computing
Center (DKRZ) the most appropriate climate models were selected for the tailored attribution study by
including the ones that had at least three initializations in the DAMIP archive and passed the validation
tests outlined below for the maximum temperature (Tmax) that is analyzed in the attribution study. The
climatology of Tmax and the spatial pattern of the yearly averaged maximum temperature were visually
compared between the models and the gridded E-OBS dataset to evaluate whether the models are able
to represent the climate conditions over Germany. Additionally, the parameters of a Generalized
Extreme Value (GEV) distribution fitted to the simulation data were compared with a fit to the E-OBS
data to check whether they agree within their uncertainty bounds. Furthermore, a general consistency
check was performed for each model ensemble. The evaluation procedure is similar to the one used in
World Weather Attribution studies (see e.g. Philip et al., 2020). Simulations of CMIP6 models that
passed the validation were further analyzed (see Table A1 for a list of the models).
The following steps are required to calculate the risk ratio: CMIP and DAMIP Tmax data from all
available initializations of the model were selected for the German region and for the 30-year timeframe
from 1985-2014. The data were averaged over the region and a 17-day running mean was calculated,
based on the event definition which is further elaborated in Sect. 3.4. The yearly block maxima were
then selected from all initializations and a GEV fit was used to estimate the probability of heatwaves in
the simulation data that are equivalent to the observed event of 2018. To account for offsets between
observed and simulated temperatures, we analyzed a simulated heat event which has – in the historical
simulations - the same probability as the observed heatwave, i.e. while the simulated event may not
reach the same temperature as was observed in 2018, the temperature threshold used to analyze the
simulations has the same return period as the observed event (see also Philip et al., 2020; Tradowsky et



al., 2022). To increase the robustness of the results a 1000-member bootstrap was used and a GEV distribution was fitted to each of these 1000 alternative time series. The probability ratios (PR) were then calculated from the probabilities of such heatwaves in the historical and hist-nat simulations using the GEV fits to the original simulation time series and to the 1000 alternative time series, according to equation (1):

$$PR = \frac{P_{historical}}{P_{hist-nat}} \tag{1}$$

$P_{historical}$ is the probability of the event to occur in the historical CMIP scenario and $P_{hist-nat}$ is the probability in the naturalized DAMIP scenario in which anthropogenic greenhouse gas emissions are fixed to pre-industrial times.

A probability ratio > 1 indicates an increase in the probability of such an event due to anthropogenic climate change, a result which is typically found for recent heatwaves (see e.g. Stott et al., 2004; Philip et al., 2021).

## 3 Results

### 3.1 Detection and description of the 2018 summer extremes

The 2018 summer was an extreme season from the climatological perspective for many regions in Europe. An intense heatwave first affected Scandinavia in mid-July and then extended towards central Europe and later Iberia, spanning a total period of four weeks. The maximum heatwave duration was seen in Scandinavian regions, reaching 20 consecutive days (Fig. 1a). Cumulative heat reached peak values in parts of Norway, Sweden, Germany, France, Ireland and the UK (Fig. 1b). The cUTCI index showed periods of extreme heat stress in Portugal and southwestern Spain, very strong heat stress in northern and central Germany, central-western Poland, large parts of France and Iberia, and strong heat stress in most of eastern Europe, Finland, southern Scandinavia and parts of the British Isles (Fig. 1c). The high intensities in Turkey and the Caucasian region were not caused by the same weather pattern as the event described in this paper and are thus not discussed here.

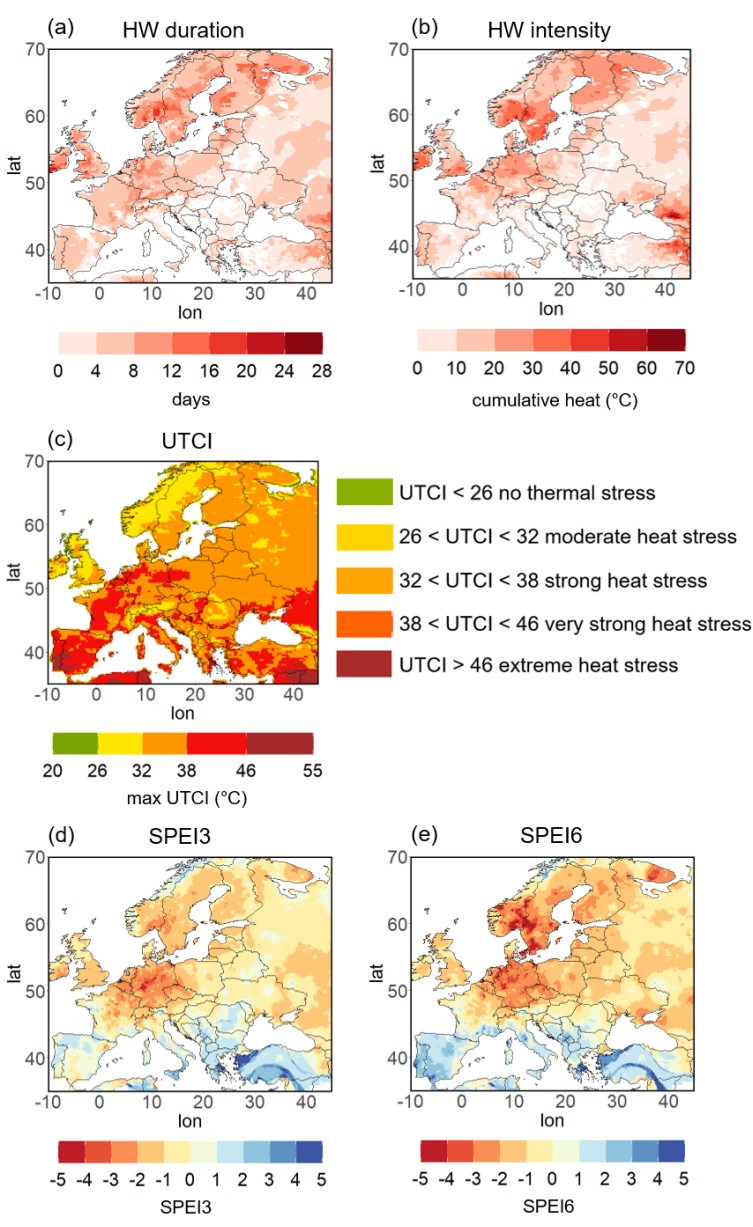

**Figure 1:** Spatial representation of European heatwave (ERA5) and drought (E-OBS) in the 2018 summer. (a) Maximum heatwave duration in days (grid point-based, exceedance of 90th percentile of daily maximum temperature). (b) Cumulative heat (in °C). (c) Maximum UTCI in the 2018 summer per grid point and respective heat stress category. (d) SPEI3 August. (e) SPEI6 August. Respective maps for SPI can be found in the Appendix (Fig. A1). Reference period used in all metrics: 1981-2010.





In northern and central Europe, the heatwave was preceded and accompanied by intense drought

conditions. As an example, the meteorological drought is depicted in terms of the SPEI3 and SPEI6
values for August (Fig. 1d,e) that were particularly low in central and northern Europe (correspondent
SPI shown in Fig. A1). The cumulative effect of low precipitation and high evapotranspiration lead to
lower values of the SPEI6 index in many European regions compared to SPEI3. The most extreme
values (SPEI6 < -5) are identified for southern Norway and Sweden. For Germany, the drought
conditions can also be seen when using the complementary approach based on climate networks
(Schädler and Breil, 2021). Figure 2a shows the spatial distribution of node degree anomalies, compared
to the reference period 1981-2010, for dry days in Germany for the 2018 summer, as a measure of
drought spatial coherence (connectivity). High anomaly values are identified for large areas in central
and northern Germany highlighting the exceptional drought, while no anomalies are found over south
Germany. The thermopluviogram for Germany depicts temperature and precipitation anomalies for
Germany, and confirms that the extended warm period of April to October 2018 was the most
exceptional in terms of precipitation deficit and heat anomaly compared to the reference period (1981-
2010) since 1881 (Fig. 2b). When considering different seasonal periods, such as March to August, or
June to August only, 2018 remains a very extreme season (see Fig. A2). In summary, while the heatwave
was most intense in southern Scandinavia, 2018 stood out as the most intense compound heat and dry
event in the observational history for Germany, in agreement with Zscheischler and Fischer (2020).

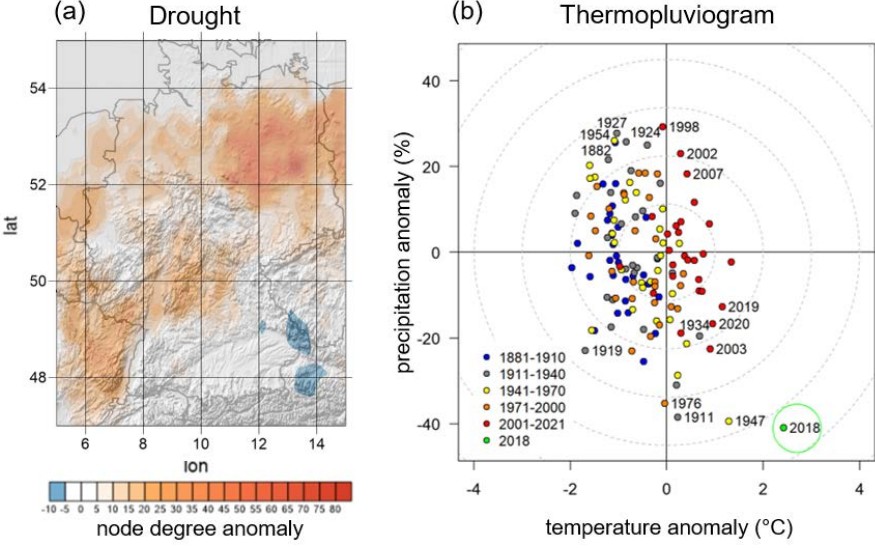


**Figure 2:** Summer 2018 heatwave and drought in Germany. (a) Climate network node degree anomaly
(as a proxy for spatial coherence) for dry days in summer (June to August; JJA) 2018 in Germany (E-
OBS data, reference period 1981-2010). (b) Thermopluviogram for the growing season, April to





October, of the years 1881-2021 for Germany showing the temperature and precipitation anomalies
from the climatological mean (DWD observational data, reference period 1981-2010). 2018 is
highlighted with light green color. Thermopluviograms for different periods can be found in the
Appendix (Fig. A2).

## 3.2 Dynamical drivers and evolution

In order to characterize large-scale circulation features for summer 2018, we used a number of different
and complementary metrics to describe the multi-faceted characteristics of the event. First, we analyzed
the blocking conditions for this season, as the occurrence of heatwaves is directly associated with
summer blocking or – for the lower latitudes in Europe – to atmospheric ridges (Woollings et al., 2018;
Sousa et al., 2018; Kautz et al., 2022). Using the blocking detection algorithm, we confirm that for the
2018 summer, blocking is detected over Great Britain from late June into the first ten days of July as
well as over Scandinavian and Ural regions for most days of July (Fig. A3). Compared to the
climatological occurrence of blocking frequency, the percentage of blocked days in June/July 2018 was
20-60 % higher in the mentioned areas (Fig. 3a,b), indicating blocking frequency values above the 90th
percentile (Fig. A4). This large-scale set up for the summer time (see e.g., Kautz et al., 2022, their
Figure 2b) leads to the development of a heatwave collocated with the center of the blocking, while
unsteady weather conditions may happen on the block edges.

The establishment of a long-lived blocking anticyclone is consistent with the development of a

double jet stream state over Eurasia, with two maxima of the zonal mean zonal wind at the 250 hPa
level, which started as early as mid-May and persisted until 25th of July, with only a few days in between
not characterized by double jets (Fig. 3c). The period 04–25 July was characterized by a continuous
persistent double jet configuration, according to the SOM-based detection scheme of jet stream states.
These 22 consecutive days of double jets make 2018 one of the longest such events in the study period
(1979–2020), the longest being that of 2003 (Rousi et al., 2022; their Figure 4). The initiation of the
heatwave in Europe happened a few days after the initiation of this persistent double jet event (see Fig.
4), highlighting a potential role of the double jet structure in preconditioning the flow and favoring the
onset of a heatwave in the region of weak winds between the two jets, where the blocking anticyclone
lies (Rousi et al., 2022). This large-scale set up typically corresponds to the occurrence of the summer
NAO+ (sNAO+) regime, as confirmed by the circulation regime approach applied on the 2018 summer.
Indeed, most of July 2018 was dominated by a sNAO+ pattern, typically characterized by a more
northerly location and smaller spatial scale than its winter counterpart (Folland et al., 2009). This is in
agreement with previous studies (e.g. Drouard et al., 2019) showing a strong positive EOF-based NAO
anomaly in this time period that is consistent with large parts of the seasonal anomalies observed during
summer 2018.



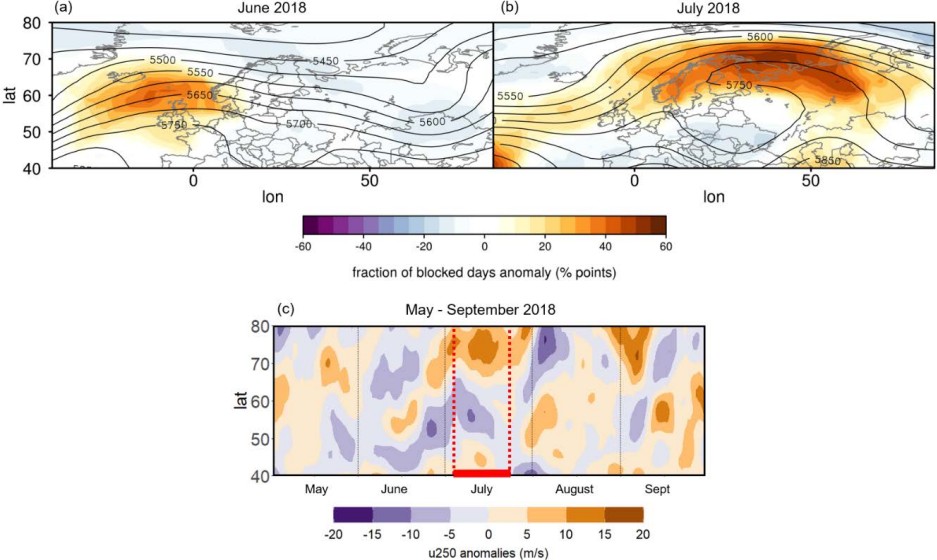

**Figure 3:** Blocking frequency anomalies for (a) June and (b) July 2018 (shading, contour lines show mean geopotential height at 500hPa plotted every 50hPa). (c) Eurasian zonal mean zonal wind at 250hPa for May-September 2018 (shading; 5day running means centered on each day from 01.05-30.09.2018). The red lines mark the duration of the longest double jet event (04-25.07.2018).

The analysis of Rossby wave activity permits the evaluation of the development of the blocking, NAO+ phase and the corresponding double jet structure for the summer 2018. Results show an eastward propagation of Rossby wave packets from the Pacific towards the Atlantic Ocean, the British Isles, and finally towards the European continent during the last 10 to 15 days of June and before the initiation of the heatwave over Scandinavia (Fig. 4). On the other hand, this was not the case for August, when the peak over Iberia occurred, which highlights the different mechanisms involved in this heatwave, rather than Rossby wave activity coming from the Pacific. Indeed, heatwaves and precipitation deficits in this location are primarily associated with amplified subtropical atmospheric ridges rather than midlatitude blocking situations (see Woollings et al., 2011; Sousa et al., 2017; 2018).

Further, a backward trajectory analysis was conducted to determine the origins of the air masses that were present during the different heatwave phases and their evolution. Three grid points were chosen to represent the three affected areas and time segments of the heatwave: one over Scandinavia (Utsjoki, Finland) initialized on 18 July 2018, one over central Europe (Bernburg, Germany) on 31 July, and one over Iberia (Alvega, Portugal) on 4 August 2018 (Fig. 5). The backward trajectories showed the remote origin of the mid-troposphere air masses, especially in the case of Utsjoki (Fig. 5a), where it primarily originated over the central North Atlantic. This is also true for the mid-troposphere air masses in the case of Bernburg (Fig. 5b). However, in the last 48 hours, descending air masses were



observed, pointing to an adiabatic warming by compression. Trajectories starting in the lowest 200 hPa
at Bernburg, indicate that air masses stemmed from a region to the south and east close to the starting
location, indicating relatively stagnant air masses as already discussed in Spensberger et al. (2020). In
the case of Alvega (Fig. 5c), air masses starting between 700 and 1000 hPa experienced several rising
and sinking motions on their way from the south and southeast (e.g. Algerian desert, Atlas Mountains,
Mediterranean Sea), towards the Iberian plateau and coastal regions, thus documenting their local to
regional origin and largely stagnant conditions (in line with Santos et al., 2015).

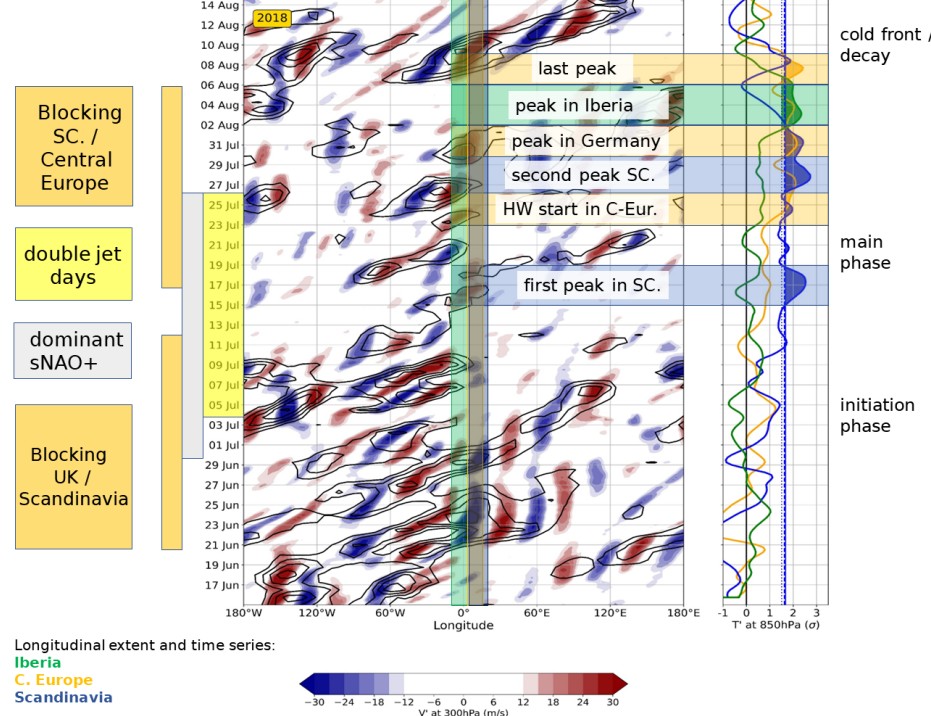

**Figure 4:** Hovmöller diagram for the period of 15.06.–15.08.2018. The longitudinal extent of three core
heatwave regions (Iberia, Central Europe, Scandinavia), as well as their temperature time series at the
850 hPa level as standardized anomalies (T') on the right, are marked in green, orange and blue,
respectively. Periods when T' was above the respective 95th percentiles are shaded. Both temperature
(T') and meridional wind at the 300 hPa level (v') are anomalies with respect to their smoothed annual
cycles. Rossby wave packet amplitude (E) is depicted in contours from 24 to 38 m/s in steps of 4 m/s,
v' as color shading from -30 to 30 m/s. Both fields are weighted by the cosine of latitude and
meridionally averaged over above-median grid points within the 40-80° N latitude band (self-adjusting,
depending on the location of the largest amplitudes). Days with a dominant positive phase of the





summer North Atlantic Oscillation (sNAO+) pattern, double jet days, and blocking days are marked on
the left.

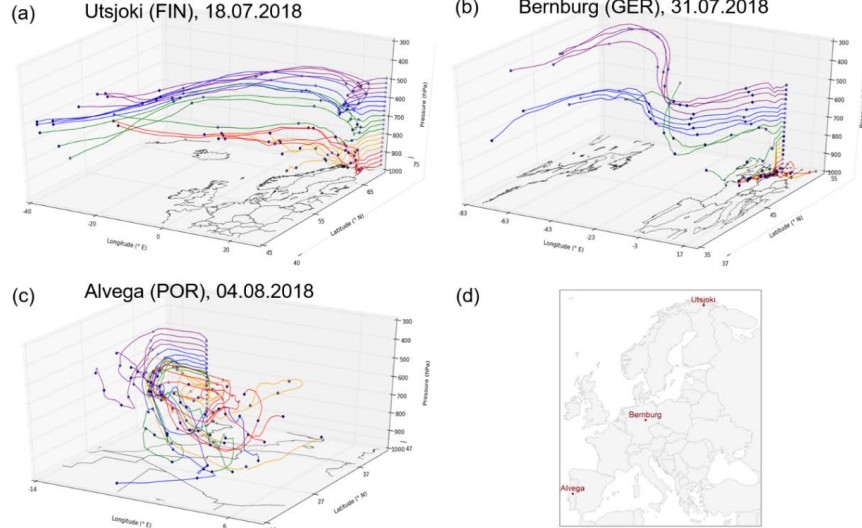


**Figure 5:** 10-day backward trajectories in 25 hPa steps between 1000 hPa and 500 hPa for the three
location coordinates. (a) Utsjoki, Finland, initialized 18.07.2018. (b) Bernburg, Germany, initialized
31.07.2018. (c)Alvega, Portugal, initialized 04.08.2018. For every 100 hPa, a different color is used for
the trajectories. Each black dot is representative of a 24-hour time step. (d) Geographical locations of
the three points.

In order to infer causal hypotheses for the existence of the Scandinavian block, the trajectory

approach was extended to obtain the origins of low potential vorticity (PV) air masses that formed the
upper-tropospheric part of the Scandinavian anticyclone (see Sect. 2.2.3). For the sake of brevity, only
maps of 7-day and 3-day trajectory density on 18 July 18 2018, around the maximum heatwave day in
Scandinavia, are shown in Figure 6, but other days corroborate the inferences below (not shown). Figure
6a shows the density of 7-day backward trajectories, indicating that air masses were steered from the
Western North Atlantic over the British Isles to Scandinavia. This is in line with the propagation of the
corresponding Rossby wave packet discussed above. Moreover, using the method described in
Zschenderlein et al. (2020, their Fig. 4), the role of a remote warm conveyer belt is suggested by
ascending, diabatically heated trajectories over the western Atlantic (not shown); PV is lowered in the
warm conveyer belt and then transported in the upper troposphere into the Scandinavian anticyclone
(termed "remote branch" by Zschenderlein et al. (2020). Interestingly, high trajectory densities over
central to eastern Europe, which also strongly ascended and were diabatically heated (not shown), point
towards an influence of moist convection observed under an upper-level trough in this area in feeding





low PV air towards the Scandinavian anticyclone. Such a "nearby branch" was also mentioned by
Zschenderlein et al. (2020) to be important for anticyclone persistence over central Europe. However,
in the 2018 case the nearby branch is located to the southeast, not to the southwest as for central Europe.
Three days before the peak of the heatwave, trajectories almost exclusively stem from this nearby
branch, now located more to the south of the Scandinavian anticyclone (Fig. 6b). Clearly, determining
causal pathways from this analysis is not possible, yet modelling studies with explicit convection could
shed more light on the role of the remote branch (warm conveyer belt over the western Atlantic) versus
the nearby branch over southeastern Europe for the establishment and maintenance of the Scandinavian
anticyclone.

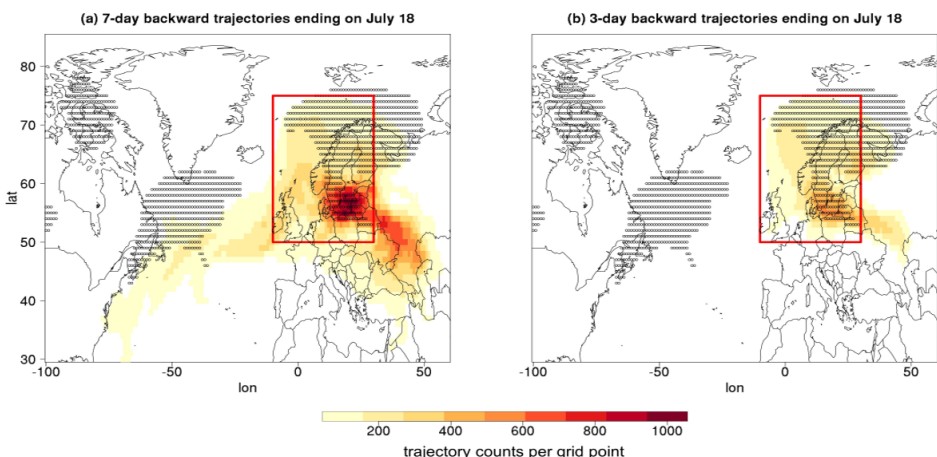


**Figure 6:** Backward trajectories for 7-days (a) and 3-days (b). Backward trajectory density maps ending
on July 18, initiated in 50 hPa steps between 150 hPa and 500 hPa for grid points within the
Scandinavian anticyclone (backward trajectories were initiated from the dotted points inside the red
rectangle; the dotted points are those defined by vertically averaged PV anomaly based on monthly
climatology < -0.7 PVU and PV < 1 PVU).
**3.3 Low-frequency precursors**
When addressing possible precursors for European heatwaves, SST anomalies over the North Atlantic
(Ossó et al., 2020) and soil moisture anomalies over continental Europe (Quesada et al., 2012) are
among the primary candidates, as outlined in the Introduction. A tripolar SST pattern with negative
anomalies over the Subpolar Gyre (SPG) was evident in spring (MAM, northern box of Fig. 7a,b). At
the same time, a pronounced precipitation deficit over Scandinavia in spring 2018 was present (Fig.
7c). The SST tripolar pattern persisted over time, with the cold SPG anomaly intensifying in summer
(JJA, Fig. 7b), and the same is true for the precipitation deficit, which increased particularly in Germany
and central Europe (Fig. 7d). The soil moisture anomaly for 2018 spring and summer (Fig. 7e,f) shows



a pattern consistent with the precipitation anomaly. LPJmL- simulated soil moisture anomalies for 2018
spring and summer (Fig. A5a,b) corroborate the spatial pattern seen in the ERA5 analysis.

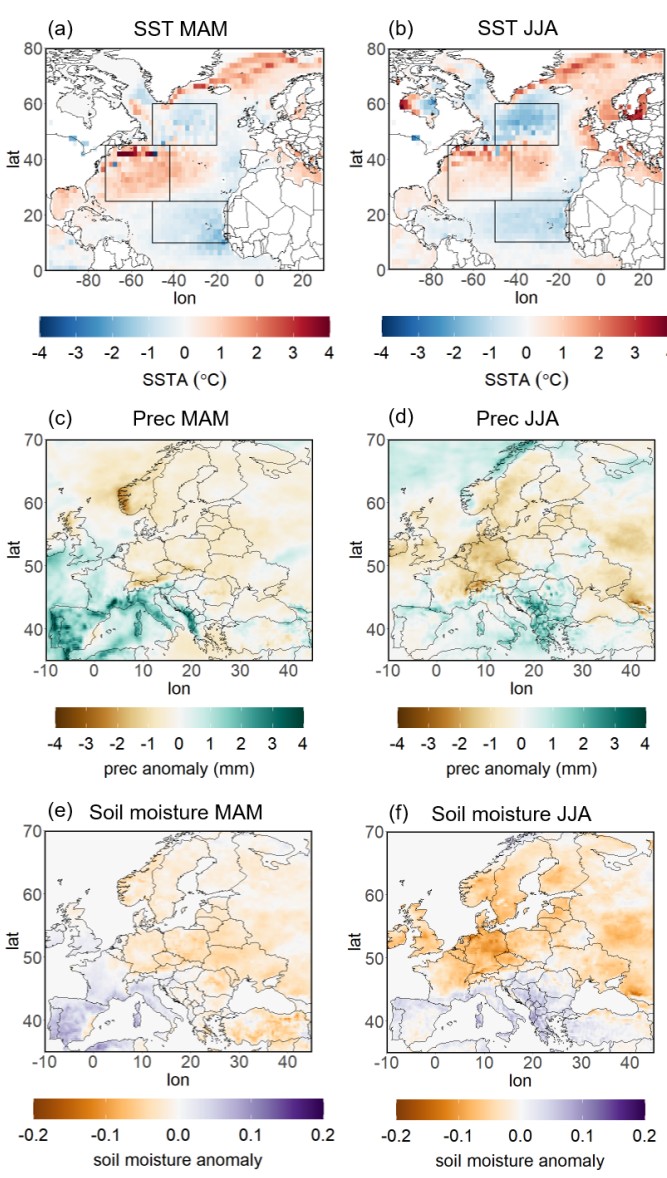


**Figure 7:** Anomalies of sea surface temperature (SST; a,b), precipitation (c,d) and soil moisture (e, f) in the ERA5 reanalysis (compared to the reference period 1981–2010) for spring (March to May; MAM; a, c, e) and summer (June to August; JJA; b, d, f) months. Boxes in (a) and (b) indicate the regions for the tripolar SST pattern.





Having established that the large-scale soil moisture anomaly is consistent with the SST and
precipitation anomalies, we investigated the temporal development of the soil moisture pattern over
Germany. Reduced soil moisture often facilitates the occurrence a summer drought and heatwaves, as
the soil moisture determinant for evapotranspiration (or lack thereof) directly links to the surface
temperature and relative humidity at the land surface. Thus, soil moisture and latent heat flux were used
to identify periods of moisture limitation (denoted by positive correlation coefficients between the two)
and wet conditions (negative correlation coefficients), under which the latent heat flux is primarily
controlled by the atmosphere. The derived (Fig. 8) and simulated (Fig. A5c,d) time series for the soil
moisture-latent heat flux correlations are based on daily data centered on 92-day running periods for
Germany. Additionally, centered 92-day running mean soil moisture is shown. The time series were
spatially averaged over all land points for northern (Fig. 8a,c; Fig. A5c,e) and southern Germany (Fig.
8b,d; Fig. A5d,f). Germany is usually not in the moisture-limited regime, but extraordinary hydrologic
conditions can lead to a shift from an energy-limited evaporative regime to moisture-limited conditions
(Lo et al., 2021), increasing the surface temperature and enhancing the sensible heat flux. The soil
moisture anomaly in March 2018 was low all over Germany (Fig. 8a,b; Fig. A5c,d; note that the LPJmL-
simulated soil moisture estimates are lower in absolute terms compared to ERA5, which is likely the
result of lower soil water holding capacity assumed in this model) and thus did not yet limit
evapotranspiration and latent heat flux. Warm conditions in spring caused a high latent heat flux all
over Germany, indicating a strong energy-limitation (Fig. 8c,d; Fig. A5e,f). High latent heat fluxes, in
turn, lead to a severe depletion of the soil moisture up to a depth of 1 m, starting at the end of March
and continuing until July in northern Germany and mid-August in southern Germany. The precipitation
deficit (Fig. 7c,d) further exacerbated the drying of the soils, and shifted the evaporative regime from
energy-limited to moisture-limited conditions. The latter prevailed between June and August 2018,
indicating that the anomalously dry soils during the 2018 summer further augmented the hot surface
temperatures (Dirmeyer et al., 2021; Orth, 2021).
In summary, the observed and modelled spring and early summer SST anomalies over the North
Atlantic and European soil moisture anomaly patterns for 2018 are in line with those identified for other
recent hot summers. Moreover, the dried-out soils and vegetation may have enhanced the maximum
temperatures by leading to anomalous latent heat fluxes.





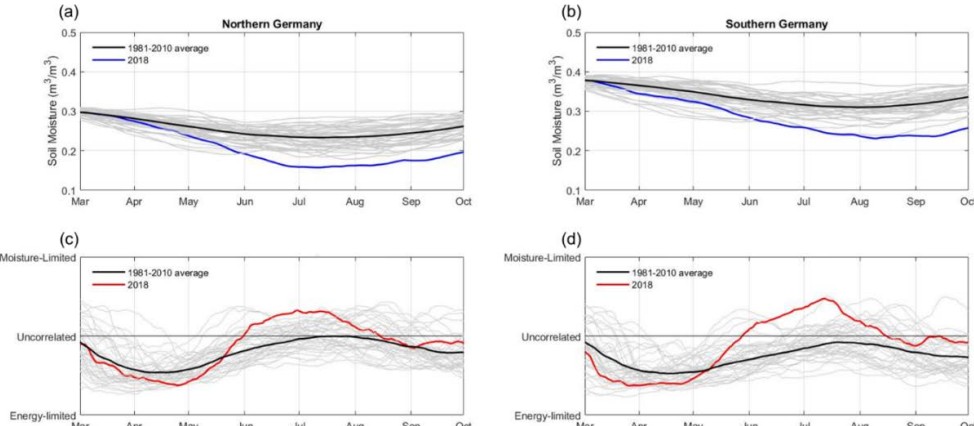

**Figure 8:** (a) Time series of centered 92-day running mean soil moisture averaged over all land points
of northern Germany (51° N – 55° N and 4° E – 16° E) for the period March-September of 1981-2020.
The grey lines denote individual years, the black line the average of 1981-2010, and the blue line 2018.
(b) As (a) but for southern Germany (48° N-51° N and 4° E – 16° E). (c) Time series of soil moisture-
latent heat flux coefficients based on 92-day running periods for the growing period covering March to
September for the years 1981-2020 for northern Germany. The grey lines denote individual years, the
black line the average of 1981-2010, and the red line 2018. Energy-limited is related to a correlation
coefficient of -1, and moisture-limited to a correlation coefficient of 1. (d) As (c) but for southern
Germany.
**3.4 Attribution of the 2018 extreme heat**
This section evaluates how anthropogenic climate change has affected the likelihood of similar
heatwaves under present climate conditions and how it will affect their likelihood at global warming
levels of +1.5° C and +2° C compared to pre-industrial times.
As defined by the cumulative heat metric, the 2018 summer was the 2nd warmest summer over
Europe following 2010, surpassed again in 2019 and 2021 (not shown), ranking it 4[th] warmest by now,
with 2022 being another candidate for warmest summer yet. In the period of 1950-2021, ERA5 data
exhibits a 7 % likelihood of 2018 cumulative heat levels (black PDF in Fig. 9). MPI-GE, which is shown
to adequately represent the variability and forced anthropogenic changes in observed temperatures
(Suarez-Gutierrez et al., 2018; 2021), is also well able to capture cumulative heat (gray PDF in Fig. 9)
as compared to ERA5. Under recent climate conditions, the 100 members of MPI-GE simulate a 9 %
likelihood of exceeding 2018 levels, making this roughly a 1-in-10-years event. This is in line with an
earlier attribution study by the World Weather Attribution (WWA) team who found return periods of
about 1-in-10-years for Scandinavia and slightly less in the Netherlands (WWA, 2018). Under stronger
global warming, this likelihood reaches 69 % in a +1.5° C world, and 96 % in a +2° C world (orange



and red PDFs in Fig. 9). Thus, conditions as extreme as the summer 2018 are projected to occur two
out of every three summers in a 1.5° C warmer world, while given 2° C of global warming they occur
virtually every single summer. The extreme summer 2018 represents a fairly average summer in a +1.5°
C world. In a 2° C warmer world, the cumulative heat during the average summer is twice as large as
the 2018 levels, while the most extreme 2° C world summers could exhibit more than four times more
excess heat compared to the recent climate conditions.

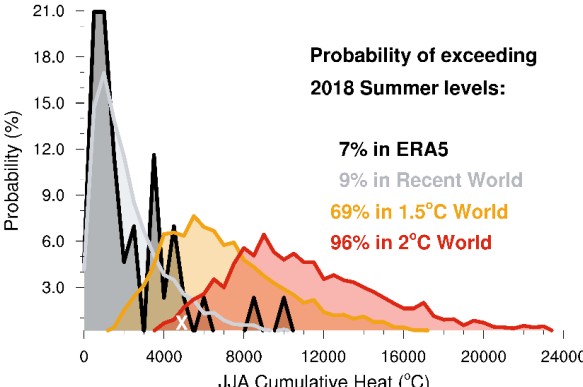

**Figure 9:** European ERA5 (1979-2021; black) cumulative heat versus MPI-GE under recent (1979-
2021; gray), future +1.5° C (2020-2049; orange), and +2° C (2050-2079; red) compared to pre-industrial
times warmer worlds. 2018 summer from ERA5 data is marked with white X. Daily maximum
temperatures (Tmax) for summer months (June to August; JJA) over land grid points only. Anomalies
with respect to 1981-2010. ERA5 data regridded to coarser resolution of MPI-GE. Probabilities are
normalized to percentages (divided by total number of years in period). Bin size is 500° C.
To estimate how much more likely the heat event of 2018 has become in Germany in recent decades

due to anthropogenic climate change, its probability ratio was calculated based on historical and hist-
nat (pre-industrial-type) simulations from the CMIP6 archive. In a first step, we defined the extreme
event for which the tailored attribution analysis for Germany was conducted. We analyzed the
maximum daily temperature (Tmax) averaged for a box over Germany (47.5-55° N, 6-15° E) and to
account for the prolonged heat of 2018, we used the Tmax as a spatial average over seventeen days
(Tmax17). This length was defined based on the longest period of consecutive days with Tmax above
30 °C in German weather stations on record. Using this length, resulted in the longest return period.
Thus, annual block maxima of this variable (Tmax17) were constructed within the GEV fit and the
return periods were calculated. The return period of the 2018 summer Tmax17 (approximately 31° C
in E-OBS) was estimated as 108 years, making it a heatwave that is expected less than once in a lifetime
and can therefore have considerable impacts. It should be acknowledged that such a return period



estimate contains uncertainties as the time series used to calculate it are shorter (about 70 years).
Following the analysis of observation-based data, the following models were analyzed: CanESM5,
CNRM-CM6-1, ACCESS-ESM1-5, IPSL-CM6A-LR, HadGEM3-GC31-LL, and MRI-ESM2-0 (see
Table A1 for further details on the models used).

The probability ratio of the 2018 summer heatwave occurrence in Germany is shown for all

analyzed models in Figure 10. For all models the probability ratio estimated on the original simulation
data is larger than 1, meaning that the probability of such a heatwave has increased due to anthropogenic
climate change. The red bars provide uncertainty ranges bases on the 1000 bootstraps. The best estimate
in all analysed CMIP6 models (black squares) is > 2, again in line with the WWA findings despite a
rather different event definition (WWA, 2018). For readability of the results, the x-axis in Figure 10 is
only extended to a value of 100 with larger values omitted due to the large uncertainties. In fact, the
upper range of probability ratios for some models is invalid as the event had a zero probability of
occurrence in the hist-nat scenario, indicating that such an extended heatwave would have been very
improbable under pre-industrial conditions.

In summary, the analysis of the impact of anthropogenic climate change on the heatwave in summer

2018 shows that such heat events have already become more frequent, i.e. their probability has increased
compared to pre-industrial conditions. Furthermore, it is expected that such heat events will become
even more likely in a warmer world.

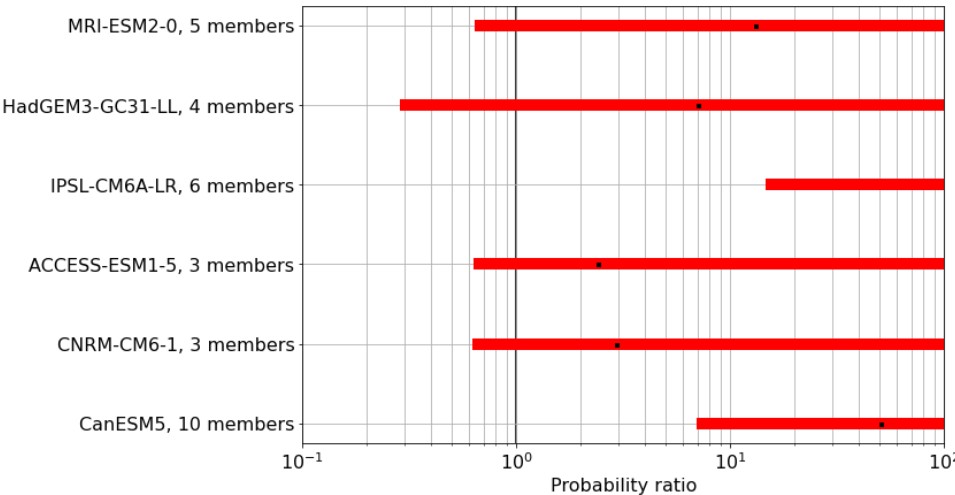


**Figure 10:** Probability ratio (PR) of the 2018 summer heatwave occurrence in Germany in the analyzed
CMIP6 models (see Table A1). The black squares show the PR estimated based on the original
simulation time series and the red bars show the $5^{th}$ to $95^{th}$ PR percentiles calculated from a 1000-
member bootstrap. The number of available DAMIP ensemble members is given together with the





606 model name and the originating institution on the y-axis. The vertical thick black line indicates a PR=1,

607 above which the likelihood of such an event has increased compared to pre-industrial times.


609  Drought attribution is notoriously difficult due to the fact that global models only crudely reproduce

610 convective precipitation, which is the main mode of rainfall in summer. While evapotranspiration is

611 increasing with warming, the question whether or not this can be compensated by stronger downpours

612 to avoid hydrological (or agricultural) drought cannot be answered with any degree of certainty at the

613 moment. Drought episodes are expected to increase (Masson-Delmotte et al. IPCC, 2021) across the

614 world, but the frequency of occurrence and the actual change in risk cannot be quantified yet.

615 Nevertheless, it is likely that the prolonged 2018 drought, followed by two more below-average rainfall

616 years in 2019 and 2020 in Germany, is partially attributable to human-induced climate change. Given

617 that attributable global warming is approximately 1.1° C (2011-2020), corresponding to 100% of the

618 observed warming and warming over land is much more rapid, Europe has already warmed by ~2°C,

619 with summer warming being particularly amplified due to soil moisture feedbacks with increased

620 sensible heat flux. Together with the potential dynamic feedback discussed above, the average summer

621 Tmax in Europe may well exceed 3° C above pre-industrial conditions already. This is corroborated by

622 a recent WWA study which analyzed the recent UK heat record and found that climate change added

623 4° C to the observed record Tmax. What used to be a 36°C day is now a 40°C day (World Weather

624 Attribution (WWA), 2022).

## 4 Discussion and conclusions

626  The extreme heat and drought of the summer 2018 has been studied from a multi-faceted weather

627 and climate perspective. We looked at hot and dry summers over Europe using different analysis

628 approaches to study the extremeness and attribution to anthropogenic climate change (climate

629 perspective), as well as synoptic dynamics in concert with slowly varying boundary conditions at the

630 ocean and continental surfaces (seasonal and weather perspective). The 2018 summer is found to be a

631 unique historical example of persistent heatwave and drought conditions in large parts of Europe. This

632 is particularly true for northern and central Europe, regions which - unlike the seasonal drought in the

633 Mediterranean- are historically not so accustomed to this kind of concurrent hot and dry summer

634 extremes. The 2018 summer is one more case in a cluster of intense heatwaves facing Europe over the

635 last decades (Russo et al., 2015; Becker et al., 2022). The 2018 drought was an intense, large-scale

636 event, promoting strong land-atmosphere coupling that exacerbated the heatwave (Dirmeyer et al.,

637 2021).

638  Regarding the large-scale atmospheric conditions conducive of the summer 2018 extremes, we

639 provided detailed evidence on the blocking anticyclones, persistent double jet stream configurations,

640 sNAO+ phase, Rossby wave activity, and different air mass origins. For example, the persistent double



jet stream event, combined with record high positive sNAO (Drouard et al., 2019), seems to have played
a role in the long duration of the 2018 heatwave.  Additionally, according to Li et al. (2020), the
collaborative (not mutually exclusive) roles of sNAO+ and European blocking could favor the
frequency, persistence, and magnitude of heatwaves over Europe, as the sNAO+ related blocking events
are quasi-stationary and more persistent compared to the non-NAO+ related ones. Evidence is provided
regarding the origin of the low PV air masses in the upper-tropospheric blocking anticyclone over
Scandinavia; while in its initiation phase, backward trajectory analyses point to a role of a western North
Atlantic warm conveyor belt, we provide hints that its maintenance could be supported by low PV air
stemming from moist convection in the trough flanking the block to its southeast, i.e. over Eastern
Europe. However, further analysis is needed to address the direction of causality behind this link. On
the other hand, our analysis suggests that the later heatwave phase over Iberia has different drivers, as
the air masses originated locally or were advected from nearby areas (e.g. North Africa) and are not
necessarily directly associated with the propagation and breaking of large-scale Rossby waves as over
Scandinavia (Santos et al., 2015; Sousa et al., 2019).

The dominant oceanic and large-scale conditions of the North Atlantic might have supported the

development of the 2018 heatwave. The physical reasoning on the relationship between the North
Atlantic SST tripole and exceptionally cold North Atlantic ocean, the jet stream set up and the
occurrence of the heat wave was proposed by Duchez et al. (2016) based on the summer 2015 event.
Here, we documented that similar anomalies were also observed during the spring of 2018. While the
atmospheric forcing is associated with the anomalous jet stream positions and blocking, they in turn
influence the precipitation patterns over Europe, leading to changes in the soil moisture content.
Although such a process enhances the potential for a heat extreme, the meteorological factors are the
ones that determine the timing and duration of the heatwave. Dedicated modeling experiments and
causal inference algorithms will be key to test the hypothesis of a causal link between spring North
Atlantic SSTs and subsequent summer extremes in Europe. Moreover, the patterns of North Atlantic
SSTs are acting on top of the warming background climate, which may further modify the type or the
magnitude of those relationships (McCarthy et al., 2019).

The severe soil moisture depletion in Germany between April and July of 2018 reflected the

persistently warm and dry conditions and led to anomalously dry soils in summer. The drought
conditions in the soil pushed its state into the transition zone conditions, in which soil wetness plays a
direct role in influencing the climate by reducing the evaporative cooling effect at the land surface and
thus enhancing hot and dry conditions. The moisture-limited conditions that prevailed between June
and August 2018 indicated that the hot surface temperatures are directly linked to anomalously dry soils
during the 2018 summer period (Dirmeyer et al., 2021; Orth, 2021).

We also showed that summer 2018 was extreme in the observational record for Europe and that

heat anomalies of this magnitude are expected to occur much more often in a warmer world, being
reached up to almost every year with global warming of +2° C. Wehrli et al. (2020) provided evidence



that the anthropogenic background warming was a strong contributor to the 2018 summer heatwave in
the Northern Hemisphere, highlighting that future extremes under similar atmospheric circulation
conditions at higher levels of global warming would reach dangerous levels. Our tailored attribution
study, which analyzed how the maximum temperature, averaged over 17 days over Germany, has been
impacted by anthropogenic climate change, showed that the probability of such a prolonged heat event
has increased in all CMIP6 models analyzed here. Attribution studies that analyzed the summer 2018
heatwave in other areas of Europe also found an increase in its likelihood under anthropogenic climate
change (McCarthy et al., 2019; Vogel et al., 2019; Leach et al., 2020).
We have presented a comprehensive study of the extreme hot and dry 2018 summer in Europe,
investigating its emergence and evolution with a combination of conventional and more sophisticated
metrics and methods, with an emphasis on their synoptic-scale atmospheric drivers and a reference to
their potential precursors in spring. Moreover, by assessing the event from a climate perspective, we
provided evidence that anomalous summers of such extremity have already, and will further, become
much more frequent in a warming world. Overall, this study highlights the added value of multi-faceted
approaches for the analysis of such extreme events, and that collaboration among different fields is
crucial both for the process understanding and the subsequent quantification of impacts. At the time of
writing in 2022, yet another, potentially more extreme hot and dry summer is affecting Europe
corroborating the approach of this study, but also emphasizing the need to carry out multi-disciplinary
impact studies.
**Appendix**

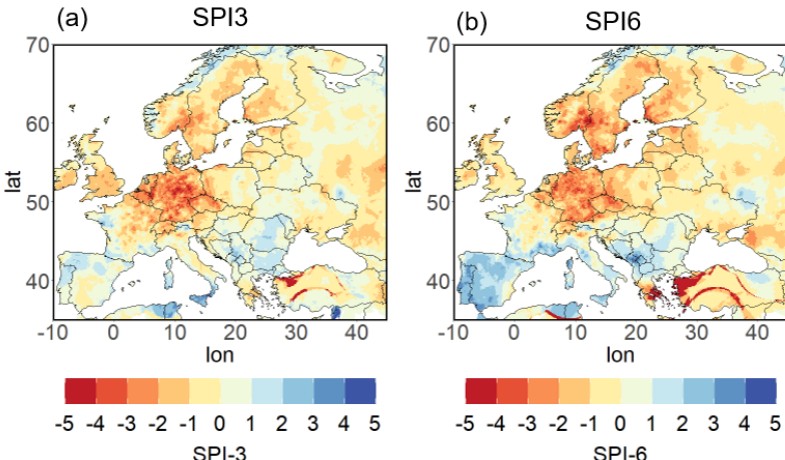


**Figure A1:** SPI3 and SPI6 August 2018 (E-OBS data, reference period 1981-2010).



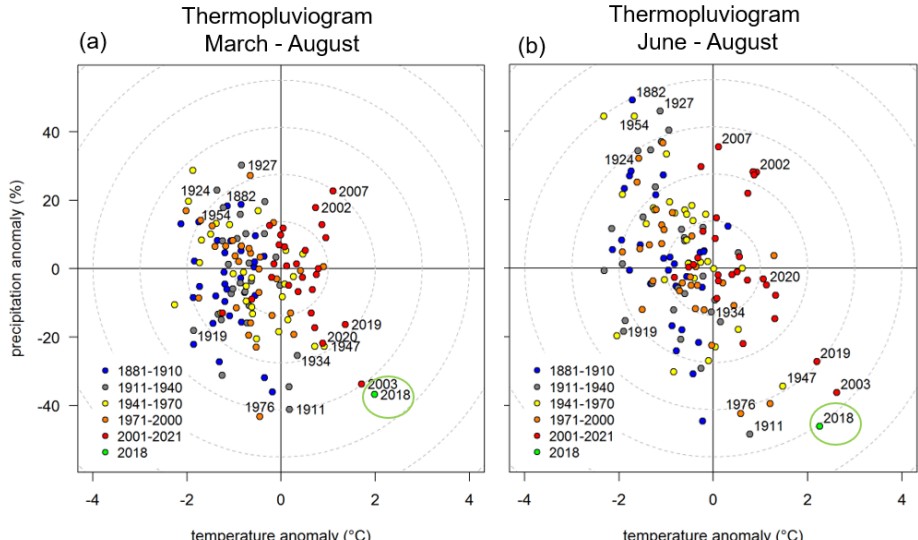

**Figure A2:** Thermopluviogram for Germany for the March – August (a) and for June-August (c). Values of temperature and precipitation anomalies from the climatological mean shown for 30-year periods of 1881-2021 (reference period 1981-2010). 2018 is highlighted with light green color.

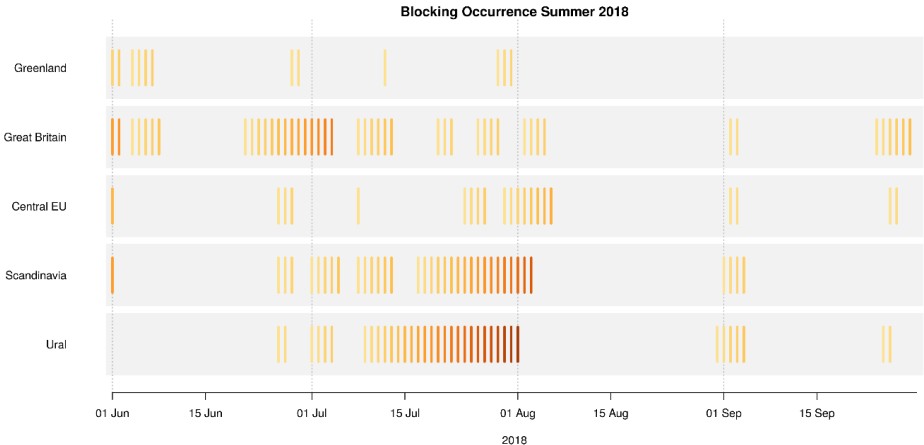

**Figure A3:** Daily atmospheric blocking occurrence and duration of consecutive blocked days (colored bars; increasing from orange to red) in different European regions from June to September 2018 in ERA5 reanalysis data. A day is defined as blocked if an area of at least 1 million km² of the specific region is blocked based on the 2-dimensional blocking index described in 2.2.3.

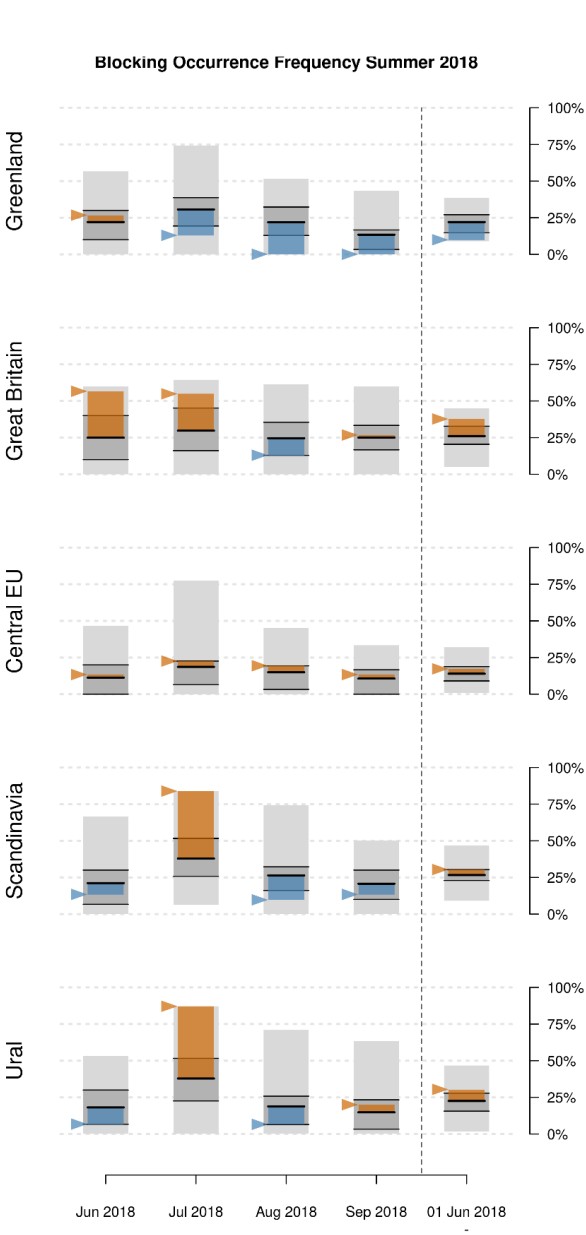

**Figure A4:** Monthly regional blocking frequency (fraction of blocked days) from June to September 2018 (colored bars and arrows) compared to climatological blocking frequencies from 1950 to 2020 in ERA5 reanalysis data. Black horizontal lines indicate the mean, light (dark) gray bars the minimum and maximum (25% and 75% quantiles) of historical blocking frequencies. Derived blocking frequencies are based on the definition of blocked days given in Figure A3.

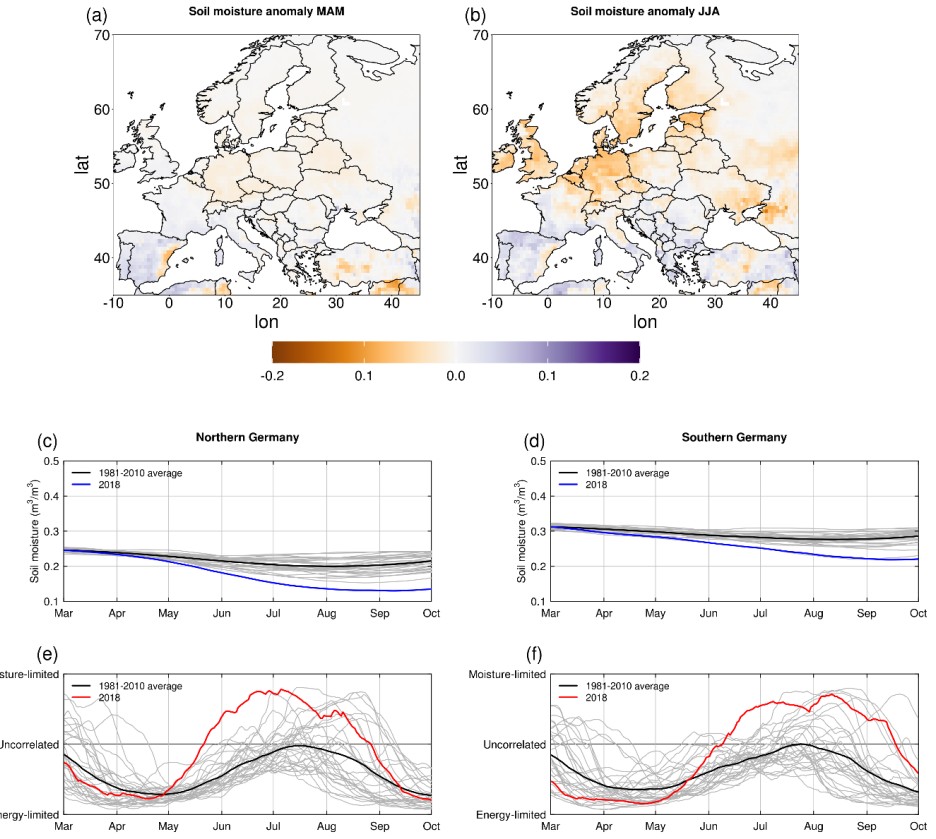

718
719

**Figure A5:** Soil moisture as simulated by LPJmL with bias-adjusted ERA5 climate forcing. Anomalies
of soil moisture for (a) March to May (MAM) and (b) June to August (JJA) as compared to the reference
period of 1981–2010. Time series of centered 92-day running mean soil moisture averaged over all land
points of (c) northern Germany (51 °N – 55 °N and 4 °E – 16 °E) and (d) southern Germany (48 °N-51
°N and 4 °E – 16 °E) for the growing period March-September of 1981-2020. The grey lines denote
individual years, the black line the average of 1981-2010, and the blue line 2018. Time series of soil
moisture-latent heat flux coefficients based on 92-day running periods for the growing period covering
March to September for the years 1981-2020 for (e) northern Germany and (f) southern Germany; the
grey lines denote individual years, the black line the average of 1981-2010, and the red line 2018.
Energy-limited is related to a correlation coefficient of -1, and moisture-limited to a correlation
coefficient of 1.


**Table A1:** CMIP6 models used for the heatwave attribution over Germany. CMIP source_id,
institution_id, data versions for the historical and hist-nat simulations, and data citation for the
CMIP6/DAMIP simulations used in the attribution study. Where initializations of the same model were




using more than one model identifying version, all of them are given and the one that was used for most
initializations is marked in bold, where possible.

| source_id | institution_id | Versions historical / version hist-nat | Versions hist-nat | Data citation historical | Data citation hist-nat |
|---|---|---|---|---|---|
| MRI-ESM2-0 | MRI | **v20190603** v20200327, v20201029 | **v20190603** v20200415 | Yukimoto et al., 2019 | Yukimoto et al., 2019b |
| HadGEM3-GC31-LL | MOHC NERC 2016 | **v20190624** v20190626 | v20190726, v20190729, v20190730, v20190805 | Ridley et al., 2019 | Jones, 2019 |
| IPSL-CM6A-LR | IPSL | **v20190614** v20190802 | v20190614 | Boucher et al., 2018 | Boucher et al., 2018b |
| ACCESS-ESM1-5 | CSIRO | v20191115, v20191128, v20191203, v20200529, v20200601, v20200605, **v20200803** | v20200615 | Ziehn et al., 2019 | Ziehn et al., 2020 |
| CNRM-CM6-1 | CNRM-CERFACS | v20180917, v20181126, v20190125, **v20191004**, v20200529 | v20190308 | Voldoire, 2018 | Voldoire, 2019 |
| CanESM5 | CCCma | v20190429 / v20190429 | v20190429 | Swart et al., 2019 | Swart et al., 2019b |

**Code availability**
Code is available from the authors upon request.
**Data availability**
The source of all datasets is listed in text and references. Further information can be made available
upon request to the corresponding authors.
**Competing interests**
The authors declare no competing interests.
**Author Contribution**
ER and AF coordinated the inter-disciplinary task force on heat and drought within ClimXtreme and
this collaborative paper; ER did the jet stream analysis, prepared Figures 1, 3c, 5d, 7, A1 and curated



most of the final figures with contributions from different co-authors (see below), and wrote the first
draft of the paper with contributions from different co-authors; FB calculated UTCI, did the Rossby
wave packet and the trajectories analysis and prepared Figures 4, 5, 6; GBA calculated the cumulate
heat metric in ERA5 and the SST anomalies; DP calculated SPEI and SPI; MB did the regional climate
network analysis and prepared Figure 2a; DN and SS prepared Figure 2b and A2; AR did the blocking
analysis and prepared Figures 3a,b, A3, and A4; JR did the weather regime analysis; LJ calculated
precipitation and soil moisture anomalies in ERA5 and prepared Figure 8; LSA and JH modeled soil
moisture with LPJmL and prepared Figure A5; LSG did the MPI-GE attribution study and prepared
Figure 9; JT did the CMIP6 attribution study and prepared Figure 10 and Table A1; GC contributed to
the CMIP6 attribution study. All authors followed the analysis from the beginning, contributed text and
edited/commented the final version of the manuscript.

### Acknowledgements

This paper is a collaborative effort within the BMBF ClimXtreme project, for which the authors
acknowledge funding (grant numbers 01LP1901A, 01LP1901C, 01LP191D, 01LP1901E, 01LP1902F,
01LP1903J, 01LP1902D, 01LP1902N, 01LP1903C, 01LP1902B, 01LP1904A). AD is supported by A4
(Aigéin, Aeráid, agus athrú Atlantaigh), funded by the Marine Institute (grant PBA/CC/18/01). EX
acknowledges support by the H2020 Project CLINT, the Academy of Athens and the Greek "National
Research Network on Climate Change and its Impact" (200/937). GF acknowledges the support of the
German Research Foundation (DFG; project no. 445572993). JGP thanks the AXA research fund for
support. We acknowledge the World Climate Research Programme, which, through its Working Group
on Coupled Modelling, coordinated and promoted CMIP6. We thank the climate modeling groups for
producing and making available their model output, the Earth System Grid Federation (ESGF) for
archiving the data and providing access, and the multiple funding agencies who support CMIP6 and
ESGF. We acknowledge the E-OBS dataset from the EU-FP6 project UERRA (http://www.uerra.eu)
and the Copernicus Climate Change Service, and the data providers in the ECA&D project
(https://www.ecad.eu).

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
