# Peer review of "The extremely hot and dry 2018 summer in central and"

_EGUsphere, 2022_

## Author Comment (AC1)

General response

We thank the two anonymous reviewers for reading our article carefully and providing constructive criticism. We have done further work to account for their suggestions and to address their concerns. In summary, we reduced the length of the paper removing some, not strictly necessary, parts; we highlighted the advance of our paper compared to recent literature on the same case study; and we incorporated the reviewers' suggestions to improve clarity in certain points. We believe that the Reviewer's comments substantially improved the manuscript. The detailed responses are provided in the two attached pdf files. The reviewers' comments appear in black font and our responses in red. All line numbers in the response documents refer to the lines in the revised manuscript without Track Changes (the manuscript is also provided with Track Changes).

Point-to-point response to Reviewer #1

The purpose of this article is to present the multi-aspect study led by the ClimXtreme research network of a dry and hot compound event: the European summer of 2018, with a special focus on Germany. The authors succeed in showing the importance of such a multi-faceted analysis in understanding the drivers and dynamics of such dry and hot compound events. The science presented here is sound, and the article is well written.

We thank Reviewer #1 for their positive and constructive feedback.

However, I believe that the authors should have presented another case study (even if more local). Indeed, this is a long and dense article, as the authors look at different aspects of this compound event, for a case study that has already been intensively documented on different aspects by various publications: exceptionality (e.g. NOAA Global climate report and Met Office report for summer 2018), predictability and drivers (e.g. Dunstone et al. 2019; McCarthy et al., 2019), drought (e.g. Toreti et al. 2019; Peters et al. 2020), dynamics (e.g. Kornhuber et al. 2019; Drouard et al. 2019; Sousa et al. 2019; Li et al., 2020; Spensberger et al. 2020), attribution (e.g. World Weather Attribution Project, 2018; Vogel et al., 2019, Hari et al., 2020). In addition, it has already been regarded as a compound event by previous studies (e.g. Bastos et al., 2021). As the authors point out along the text, most of the findings shown by the authors were already evidenced in previous scientific papers. I understand the argument of the authors on the exceptionality of such a large-scale, persistent, and intense compound event, but I find the paper very long and dense and to me it does not show any substantial new insight on this widely documented compound event. The authors should reduce the length of the paper or better justify the necessity of such a study for the 2018 compound dry and hot event and its added value.

Although we agree that the case of the 2018 European summer climate extremes has already been extensively studied, we would like to further highlight why we chose it for our collaborative paper, and which are the new aspects covered in our study compared to previous ones. Summer 2018 was an extraordinary season for Europe, with many heat and drought records broken in multiple countries and disproportional impacts affecting several sectors over extended areas. In particular, Germany was heavily affected. Here, we used multiple lines of evidence to showcase and explain the exceptionality of the 2018 extremes, ranging from the dynamical evolution of the atmospheric circulation, the co-existence of favourable precursors in the North Atlantic SSTs and the soil moisture state over Europe in general and Germany in particular, the temperature and precipitation records from long-standing weather stations around the country, and the use of two different modelling approaches, a probabilistic one based on a grand ensemble, and a multi-model one focusing on Germany to elucidate the role of anthropogenic warming in the occurrence probability of the heatwave. The April-October period of 2022 in Germany was also an extremely hot and dry one, but it was still less extreme than 2018, as seen in

the updated thermopluviogram below (Figure R1, the thermopluviograms were also updated to include 2022 in Figure 2 and A1 in the revised manuscript). In more detail, some of the novel parts of this work are:

- Regarding the dynamical mechanisms, we examined the origin of the air masses for different locations within the three broader regions that were affected by the 2018 summer heatwave (see Figure 5), Iberian peninsula (represented by a grid point over Portugal), central Europe (grid point over Germany), and Scandinavia (grid point over Finland). Further, we provided evidence that for the heatwave peak over the Iberian peninsula in August 2018, Rossby wave activity propagating from the Pacific and via the Atlantic was not one of the driving mechanisms (see Figure 4), but rather local-to-regional advection of air masses originating from Northern Africa (see Figure 5c) were determinant. Therefore, this work is spatially more exhaustive, compared to Spensberger et al. 2020 and Kornhuber et al. 2019, for instance.

- Further, we presented a tailored attribution study, particularly designed for the heat extremes during the 2018 summer heatwave in Germany, incorporating the length of the heatwave in the region, and selecting the CMIP6 models that could better capture these extremes.

Therefore, we believe that this work adds useful pieces to the puzzle of the 2018 European hot and dry extremes and contributes to their better understanding. In conclusion, we have further highlighted these points in different parts of the revised version of the manuscript, and we have reduced the length of the paper to make it more precise.

[Figure]

Figure R1. Thermopluviogram for Germany for the growing season April-October, 1881-2022. Year 2018 is highlighted with a green circle.  (This updated plot is included in the revised manuscript as Fig. 2)

Additional specific comments:

· The data sub-section is very dense and long. Maybe it would benefit of an italic title for each paragraph to quickly find the piece of information needed.

Thank you for the suggestion, which we followed in the revised manuscript. We also removed the description of the dynamical vegetation model, as we removed this part from the analysis according to the suggestion. The data subsection is shorter and easier to navigate now.

· Lines 190-197: the computation of the 90$^{th}$ percentile differs for the two heatwave indices?

The paragraph has been rephrased in order to clarify the similar definitions of the heat wave indices and metrics (lines 188-193). The computation of both 90$^{th}$ percentiles is based on daily maximum temperature, and daily maximum UTCI, for each day of the year. Our heatwave intensities are cumulative intensity measures (cumulative heat and cumulative UTCI, here in short called cUTCI) that take into account all heatwave days within the stated period.

Sub-section 2.2.2: I am not sure that three drought indicators are necessary as the purpose of the study is not to compare drought indices or to deeply evaluate the 2018 drought. Mentioning and showing only the SPEI is sufficient to evidence the occurrence and intensity of the drought, as the SPEI is a widely used drought index that considers both precipitation and evapo-transpiration. The SPI is not shown in the main body and does not bring any additional information. And the results shown with the climate network approach could be shown in another way with the SPEI. Showing only one drought indicator would contribute to lighten the text.

We thank the reviewer for their feedback on the drought indicators. We agree with the comment and given there is no much further insight offered by either the SPI, or the network indicator, compared to the SPEI, we have removed these two to make the text lighter and shorter.

Lines 230-232: The way the blocking index is computed is not clear. Is it a "hybrid" index that looks for an inversion of the Z500 meridional gradient and a strong Z500 anomaly? Are there any spatial and temporal constraints?

Thank you for the good remark. Indeed, the used blocking index is a kind of a "hybrid" index. First, the daily inversion of the meridional Z500 gradient is determined based on a modified version of the index from Scherrer et al. (2006). In addition, an associated area with strong Z500 anomalies (above 1 standard deviation) of at least $1.5{\times}10^6\,\mathrm{km}^2$ is selected. Both conditions must be met. A subsequent tracking algorithm is applied to daily blocked areas to select blocking events with a duration of at least 4 days. The method is described in more detail in Schuster et al. (2019). In addition to the "non-hybrid" index described in Schuster et al. (2019), here we expanded the definition to a hybrid index by selecting strong Z500 anomalies. We have clarified this part in the revised document (see lines 219-223).

References

Scherrer, S. C., Croci-Maspoli, M., Schwierz, C. and Appenzeller, C.: Two-dimensional indices of 1020 atmospheric blocking and their statistical relationship with winter climate patterns in the Euro-1021 Atlantic region, Int. J. Climatol., 26(2), 233–249, doi:10.1002/JOC.1250, 2006.

Schuster, M., Grieger, J., Richling, A., Schartner, T., Illing, S., Kadow, C., Müller, W. A., Pohlmann, 1029 H., Pfahl, S. and Ulbrich, U.: Improvement in the decadal prediction skill of the North Atlantic 1030 extratropical winter circulation through increased model resolution, Earth Syst. Dyn., 10(4), 901–917, 1031 doi:10.5194/ESD-10-901-2019, 2019.

· Line 259: are the soil moisture LPJmL simulations absolutely necessary for the paper? It would also contribute to reduce the length of the paper.

We agree with the reviewer that the LPJmL simulations are not strictly necessary for the paper, indeed. We thought that this is a good addition to the observations, as it confirms the observational evidence of the exceptionality of the 2018 drought. However, given that we wanted to reduce the length of the paper, we followed the reviewer's suggestion and removed this part in the revised version.

· Figures 1d and 1e: you could plot the SPEI value only when it is below the drought threshold.

Thank you for the suggestion. Figures 1d and 1e now only show values of SPEI below -1, which denote drought conditions (see below and revised Figure 1 in the manuscript).

[Figure]

Figure R2. (d) SPEI3 August. (e) SPEI6 August. Only SPEI values below -1 are shown, in order to highlight drought conditions. (The updated plots are included in the revised manuscript as Fig. 1d and 1e).

· Lines 364-365: could you show the thermopluviogram for the Scandinavian region as well?

The extensive station data network for Germany that was used for the thermopluviogram was provided by the German Weather Service (DWD). In the following plot (Fig. R3), you can see a thermopluviogram for Sweden for the summer months (JJA), which also highlights the exceptionality of the 2018 summer in terms of drought and heat there. However, due to limited data availability we could not do a plot for the whole Scandinavian region.

[Figure]

Figure R3. Thermopluviogram for Sweden for summer (June-July-August), 1881-2021. 2018 is highlighted. Input data from the Swedish Meteorological and Hydrological Institute (SMHI; https://www.smhi.se/en/climate/climate-indicators/climate-indicators-precipitation-1.91462).

· Figure 4: Could you show the NAO index for this season as you do for the blocking index?

We added a panel with the NAO index for May-September 2018 to Fig. 3 (Fig. 3c), shown also below. As discussed, the NAO was predominantly positive during this period:

[Figure]

Figure R4. NAO index for May-September 2018. Input data from CPC (https://ftp.cpc.ncep.noaa.gov/cwlinks/norm.daily.nao.index.b500101.current.ascii). (This plot is included in the revised manuscript as Fig. 3d)

·       Lines 487-490: You should also cite Dunstone et al. 2019: they studied the predictability of this summer season and evidenced the role of this tripole.

·       Lines 490-496: The lack of precipitation is also shown in Toreti et al. 2019.

·       Lines 504-506: Add a citation.

·       Sub-section 3.4: you should cite Vogel et al. 2019 in this sub-section

·       Lines 655-656: Cite Dunstone et al. 2019.

Thanks for all the good advice, we have added the suggested and additional relevant references in the different parts of the revised manuscript (see lines 476; 482-483; 493-496; 543-546; 648) .

Technical corrections:

·       Line 468: the closing parenthesis is missing.

·       Figure A2, line 702: should be "(b)" instead of "(c)".

Thanks for spotting those mistakes, we have now corrected them.

---

## Author Comment (AC2)

General response

We thank the two anonymous reviewers for reading our article carefully and providing constructive criticism. We have done further work to account for their suggestions and to address their concerns. In summary, we reduced the length of the paper removing some, not strictly necessary, parts; we highlighted the advance of our paper compared to recent literature on the same case study; and we incorporated the reviewers' suggestions to improve clarity in certain points. We believe that the Reviewer's comments substantially improved the manuscript. The detailed responses are provided in the two attached pdf files. The reviewers' comments appear in black font and our responses in red. All line numbers in the response documents refer to the lines in the revised manuscript without Track Changes (the manuscript is also provided with Track Changes).

Point-to-point response to Reviewer #2

**General comments:**

Firstly, I congratulate the authors for a very well organized and presented manuscript. It must be noted that it is not always easy to summarize and present such a multidisciplinary work, spanning different approaches, and the authors were able to provide this in a very clear and organized way, and furthermore, in a relatively concise way, which is also not easy. The quality of the writing is very good, thus very clear for the reader, and the text avoids being too "heavy", so being easy to follow.

The manuscript is well structured, with the Abstract and Introduction stating in a clear way the motivation and objectives of the work. The same is valid for the Methods, which are presented in a sound way, providing, as said before, a clear structure of the work flow, despite the complexity of the multidisciplinary approach.

As a consequence, I believe the manuscript is very close to a format suitable to be published. Accordingly, I have just some small comments, which could enhance some small parts/sections. Besides that, I have only a few minor comments regarding one or two less clear sentences and/or typos.

We sincerely thank the reviewer for their positive feedback on our work and for their suggestions that helped improve the accuracy of the manuscript.

**Specific comments:**

- L58-60: While I understand the idea in this sentence, I find it presented not in the best way. I would probably suggest the authors to be more specific regarding this specific event.

Thank you for the suggestion, we rephrased to make it more specific (lines 58-60):

"A probabilistic attribution assessment of the heatwave in Germany showed that such events of prolonged heat have become more likely due to anthropogenic global warming."

- L286: How did the authors estimate the impact and potential loss of confidence from regridding the ERA5 data?

Thank you for this comment. Due to the difference in resolution between models and observational/reanalysis data, regridding is a very common procedure to facilitate comparison. Here, we used the 'remapbil' function in CDO, which is commonly used for such applications. To make sure there is no significant difference between the "raw" and the regridded ERA5 data, we calculated the cumulative

heat for the 2018 summer in both cases (the point shown with a white "X" in Fig. 9) and when scaled for the different amount of grid cells, the two values are almost identical (i.e. 1.90 ℃ without regridding, and 1.92 ℃ with regridding). The values are also identical for 2010, which was the year with maximum cumulative heat in Europe (in both grids ~4.014 ℃). Therefore, we are confident that the regridding does not affect the reliability of the data in representing cumulative heat over Europe.

-L309-311: Why this specific timeframe?

We decided to use a 30-year timeframe within our attribution system in agreement with a climatological timeframe (according to the World Meteorological Organization - WMO recommendation, see for example: https://www.ncei.noaa.gov/products/wmo-climate-normals) to have a sufficient number of years available per model ensemble while keeping the length of the time series short enough to minimise the effect of a non-stationary climate system. The selection of the particular 30-year timeframe (1985-2014) was based on the CMIP data availability, with 2014 being the last year for which CMIP6 historical simulations are produced. We acknowledge that the time frame could be extended using scenario simulations from CMIP6 to the end of 2020, which is the last year for which Detection and Attribution Model Intercomparison Project (DAMIP) data are available, however, we decided to focus our analysis on the historical simulation period of CMIP6.

- Regarding feedbacks between soil moisture deficit and heatwave amplification, while the presented material and evidence is in my point of view more than sufficient for this multidisciplinary approach, I would probably appreciate some more discussion on the soil desiccation mechanisms, and the approaches/methodologies to address this relatively complex subject, which have for example been very well discussed (e.g. 2010 european HWs) in works such as the ones from Miralles et al. (2014), or Schumacher et al. (2019).

Thank you for the suggestion. We added a couple of sentences and references on the feedbacks between soil moisture deficit and heatwave amplification in the Introduction (lines 126-129) and Results (lines 493-496, 517-519).

- L430-434: I understand the local/regional description, however this may be somehow slightly misleading. I am not sure if extreme heat and temperature records in NW Iberia related with the advection of a Saharan air mass could or not be completely defined as "regional". In particular, the role of the advection of desertic air masses (associated with ridge activity) for Iberian heatwaves has been discussed in Sousa et al (2019).

With the use of "local" origin of the air masses here, we refer to the closer origin (Northern Africa) of the air masses over the Iberian peninsula, compared to the "remote" origin of the air masses over Scandinavia and central Europe, which are found over the central North Atlantic. However, we understand the reviewer's concern and we rephrased this paragraph to convey the correct message, also referring to Sousa et al. (lines 421-423).

**Minor comments:**

- L467 (and other instances): "warm conveyor belt"

Thank you for spotting this, it has now been corrected throughout the manuscript.

- Fig.6 seems a bit too stretched vertically

We corrected this.

- L547/551: I suggest adding here in brackets the period considered for ERA5 and "recent climate"

Thank you for the suggestion. We added the period for ERA5 and "recent climate", which is the same one, 1979-2021. Note that in the original version of the article, the ERA5 period was mentioned wrongly as 1950-2021 in this paragraph and this has now been corrected (lines 536, 539-540).

- L618: has warmed ~2⁰C since when / compared to?

This is compared to pre-industrial levels. We added this to the text (lines 610-612) .

- L662: please put the mentioned UK record into context (very briefly of course, dates, etc.)

Thank you for this suggestion. We assume this refers to L622, where indeed the 2022 UK record was brought up without context. We were referring to the exceptional heatwave that affected large parts of the UK on 18-19 of July 2022, and we added this to the text (lines 614-615).